# OPTIMIZED OVERSAMPLING

## ABSTRACT

Many classification problems that arise in practice feature imbalanced datasets, a regime in which a lot of machine learning (ML) models show diminished performance. To address class imbalance, techniques like undersampling and oversampling are used to improve the model's performance. In this paper, we introduce a new oversampling framework, Optimized Oversampling ($O^2$), which generates synthetic minority class points by maximizing the probability of belonging to the minority class, which is estimated by a trained classification model. We show theoretically, under mild assumptions, that the points generated by $O^2$ are more likely to belong to the minority class than those generated by other approaches. Further, we benchmark $O^2$ against state-of-the-art oversampling methods on 16 publicly available imbalanced datasets using Classification Trees (CART) and Logistic Regression (LR) for the downstream classification task. The numerical experiments show that $O^2$ has an edge over current state-of-the-art oversampling methods, which is more pronounced on CART.

## 1 INTRODUCTION

Many datasets that are used for binary classification feature the problem of class imbalance, that is when the minority class has fewer samples than the majority class. In this environment, many classification models demonstrate diminished performance. For example, in classification problems involving mortality or credit card fraud the resulting dataset can be extremely imbalanced, negatively impacting the performance of ML models Ghorbani et al. (2020); Shamsudin et al. (2020). The main problem with data imbalance is that the discrimination power for the minority class is not that strong Fotouhi et al. (2019). For example, a model that always predicts the majority class as an output can be highly accurate, however have no discriminatory power.

We can establish two broad categories of solutions to the imbalanced classification problem. The most popular one focuses on oversampling and undersampling techniques on the given data. When we oversample the dataset, we balance the number of samples in each class by generating and adding synthetic samples that belong to the minority class. With undersampling, we balance the dataset by throwing away samples that belong to the majority class. So far, most existing methods are randomly generating synthetic minority class points in neighborhoods of the ones observed. Arguably, the most popular oversampling method is the Synthetic Minority Oversampling Technique (SMOTE) Chawla et al. (2002). Let $\mathcal{N}_i$ denote the $k$ nearest neighbors of data point $\boldsymbol{x}_i$. SMOTE then generates a synthetic sample by selecting a minority class point $\boldsymbol{x}_i$ and one of its $k$ nearest neighbors $\boldsymbol{x}_j, j \in \mathcal{N}_i$ and randomly choosing a point in $[\boldsymbol{x}_i, \boldsymbol{x}_j]$. There has been extensive work on improving SMOTE. Some examples are BorderlineSMOTE Han et al. (2005), ADASYN He et al. (2008) along with more recent methods including GeometricSMOTE (GSMOTE) Douzas & Bacao (2019) and Sampling WIth the Majority Class (SWIM) Bellinger et al. (2020). Two other approaches proposed for oversampling the minority class are Localized Random Affine Shadowsampling (LoRAS) Bej et al. (2021) and Over-sampling the minority class in the feature space Pérez-Ortiz et al. (2015). Finally, we note that Deep Learning algorithms have also been used for oversampling the minority class, see Mullick et al. (2019); Engelmann & Lessmann (2021). These methods have shown very good performance and significant improvements over the baseline, which is to just use the original data. While most of them offer some theoretical guarantees, these methods do not directly optimize the probability of the generated points belonging to the minority class. The second approach of dealing with class imbalance is generally based on manipulating the ML model used for classification and includes "cost sensitive" approaches like tweaking the loss function (see for example Thai-Nghe

et al. (2010), Krawczyk et al. (2014), Sun et al. (2007)). Finally, hybrid algorithms combine both oversampling techniques and cost-sensitive learning Ganganwar (2012). Note that high-performing ensemble models, like Random Forests and Gradient Boosting models, usually do not suffer from class imbalance. However, such models lack on interpretability, which can be crucial in specific tasks, i.e., healthcare applications.

**Optimization over Randomization** Our method has been inspired by work that improves over randomized heuristics by utilizing an optimization framework. One of the first references to this approach has been on designing controlled trials with few samples Bertsimas et al. (2015). Similarly, in Mahmood et al. (2022), the authors propose a principled approach on minimizing the cost of collecting enough data to obtain a model capable of a desired performance score. Another method in this line of work is related to optimal imputation for missing data where the authors introduce an optimization approach to missing data imputation Bertsimas et al. (2017). This is related to oversampling, as we also generate synthetic data, not for some missing features, but for whole samples.

## 1.1 CONTRIBUTIONS

The main idea of $O^2$ is to generate synthetic minority class points by optimizing over the inputs of a trained classifier. More precisely, we start by training a classifier to predict the probability of a point belonging to the minority class. Then, we propose optimizing over the model input to find a new data point that maximizes the aforementioned probability. Our main contributions can be summarized as follows:

1. We develop $O^2$, a deterministic and reproducible framework for generating points that belong to the minority class with high probability, utilizing LR, SVM or CART.

2. We show that the proposed optimization problem converges to a stationary point when solved with gradient-based methods. Also, under mild assumptions, the points generated by $O^2$ are in expectation more likely to belong to the minority class than those generated by other approaches.

3. We benchmark $O^2$ against the state-of-the-art oversampling algorithms on 16 datasets, using CART and LR for the downstream classification task. The numerical experiments illustrate that $O^2$ has an edge in both CART and LR. Specifically, we find that in terms of AUC, $O^2$ is ranked first on both CART and LR, while in terms of the F1 score it is ranked first on CART and second on LR.

4. We apply $O^2$ in a case study about predicting the risk of mortality of patients with blunt spleen trauma. We empirically show that oversampling can improve both the performance and the interpretability of CART.

**Notation** We use bold faced characters such as $\boldsymbol{x}$ for vectors and bold faced capital letters such as $\boldsymbol{X}$ for matrices. We define $[n] = \{1, \ldots, n\}$. The norm of a vector refers to $\|\boldsymbol{x}\| = \sqrt{\sum_{i=1}^{n} x_i^2}$.

## 2 THE $O^2$ APPROACH

In this section, we formulate the problem of oversampling the minority class as an optimization problem. The method we propose consists of 2 steps. First, we solve a problem in order to get potential minority class points. Afterwards, we utilize a filtering step allowing us to eliminate synthetic points that are not likely to belong to the minority class.

## 2.1 METHOD DESCRIPTION

Our main approach consists of using optimization to oversample the minority class and, thereby, balance the dataset. It is applicable in both parametric and non-parametric models. In the parametric case, we have a binary classifier, which using weights $\boldsymbol{w}$, for a feature vector $\boldsymbol{z}$, predicts $g_{\boldsymbol{w}}(\boldsymbol{z})$. We assume that the classifier is already trained on the original dataset and that we have learned the weight vector $\boldsymbol{w}$. Then, we suggest maximizing the probability of getting a minority class point

over the feature vector $z$. If $g_w(z)$ represents the probability that point $z$ belongs to the minority class, then it makes sense to search for data points $z$ that maximize this probability. In this case, we obtain the following problem:

$$\max_z \quad g_w(z). \tag{1}$$

To generate $n$ synthetic points we need to solve Problem (1) $n$ times. In order to avoid getting the same points, we introduce some requirements. First, we require that the newly created points are not the same with the previously generated ones. If we have already generated the synthetic points $z_1, \ldots, z_{k-1}$, we then require that $z \neq z_i$, $i \in [k-1]$, in the problem for generating point $z_k$. Further, we require that the newly created points are not the same with the points of the minority class in the original dataset. Namely, if we have $m$ minority class points $x_1, \ldots, x_m$ in the original dataset, we require that $z \neq x_i$, $i \in [m]$. The problem that captures these requirements can be formulated as follows:

$$\begin{aligned} \max_z \quad & g_w(z) \\ \text{s.t.} \quad & \|z - z_i\| \geq \epsilon, \qquad i \in [k-1], \\ & \|z - x_i\| \geq \epsilon, \qquad i \in [m], \end{aligned} \tag{2}$$

with $\epsilon > 0$. In order to make Problem (2) amenable to gradient based optimization, we add the requirements as penalty terms in the objective. For the first requirement, we add the penalty term $\sum_{i \in [k-1]} \|z - z_i\|^2$, while for the second requirement we add the penalty term $\sum_{i \in [m]} \|z - x_i\|^2$. Finally, we also add the term $-\|z\|^2$, aiming to bind each component of $z$ and therefore avoid unbounded solutions. Overall, the problem that needs to be solved to find the $k$-th synthetic data point, assuming we have obtained all previous ones, is as follows:

$$\max_z \; g_w(z) + \lambda_1 \frac{1}{k} \sum_{i \in [k-1]} \|z - z_i\|^2 + \lambda_2 \frac{1}{m} \sum_{i \in [m]} \|z - x_i\|^2 - \lambda_3 \|z\|^2. \tag{3}$$

Problem (3) can be solved with any gradient-based algorithm. Specifically, we experimented with the L-BFGS algorithm Liu & Nocedal (1989) and with ADAM Kingma & Ba (2015). The initialization can be either a random point from $\mathcal{N}(0, 1)$, a minority class point or a synthetic point from another oversampling method. The parameters $\lambda_1, \lambda_2, \lambda_3$ are picked by the user and could be tuned by cross-validation depending on the problem at hand. Next, we have the following result regarding convergence to a stationary point of Problem (3), when solved by gradient methods. The proof is included in the Appendix.

**Theorem 1.** *Assume $g_w(z)$ has a Lipschitz continuous gradient with constant $L$. When gradient methods of the form $z_{k+1} = z + a_k d_k$, where $\{d_k\}$ is gradient related [1], are applied to (3), the limit points of $\{z_k\}$ are stationary points of the objective for many choices of step-size rules including the minimization rule, the limited minimization rule and the Armijo rule.*

As a result, we have a stronger case for the quality of synthetic points we get compared to other oversampling algorithms that are not optimization-based. Specifically, consider the case where $g_w(z)$ is an LR model. We know that the sigmoid function $\sigma(x) = \frac{1}{1 + e^{-x}}$ satisfies the Lipschitz condition with constant $L = 1$ Anil et al. (2019). Also, since $g_w(z) = \sigma(w^T z)$, by the chain rule,

$$\nabla g_w(z) = \frac{\partial}{\partial w^T z} \sigma(w^T z) \frac{\partial}{\partial z} w^T z = w \frac{\partial}{\partial w^T z} \sigma(w^T z)$$

and thus

$$\begin{aligned} \|\nabla g_w(x) - \nabla g_w(y)\| &= \|w \sigma'(w^T x) - w \sigma'(w^T y)\| \\ &\leq |\sigma(w^T x) - \sigma(w^T y)| \|w\| \\ &\leq |w^T x - w^T y| \|w\| \\ &\leq \|w\|^2 \|x - y\| \end{aligned}$$

where in the latter we used the Cauchy-Schwarz inequality. As a result, $L = \|w\|^2$ and we can apply Theorem 1. In addition to local convergence, the L-BFGS method has provable global convergence when we have Lipschitz continuous gradient as well as bounded level sets Li & Fukushima (2001). We next illustrate the benefit of our approach over SMOTE variants that are based on randomization.

---

[1] A direction sequence $\{d_k\}$ is gradient related to $\{z_k\}$ if for any subsequence $\{z_k\}_{k \in \mathcal{K}}$ that converges to a nonstationary point, the corresponding subsequence $\{d_k\}_{k \in \mathcal{K}}$ is bounded and satisfies $\limsup_{k \to \infty, k \in \mathcal{K}} \nabla f(z_k)^T d_k < 0$ Bertsekas (1997).

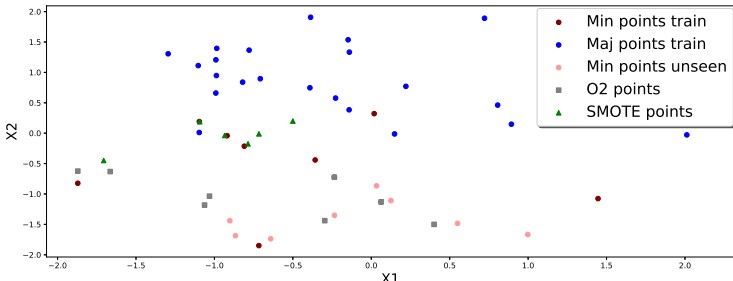

Figure 1: Oversampled points with synthetic data.

**Theorem 2.** *Consider a finite sample of $k$ minority class points obtained from $O^2$ and another oversampling method. Assume that the misclassification rate of $g_{\boldsymbol{w}}$ is $1 - p$ at the $O^2$ points and $1 - \overline{p}$ at the other points with $p > 1/2$ and $p \geq \overline{p}$. Let $y_z^j, y_r^j$ denote the true label of the $j$-th synthetic point generated by $O^2$ and the other oversampling method, respectively. Then, the sample generated by $O^2$ is expected to contain more minority class points, that is,*

$$\mathbb{E}\left[\sum_{j=1}^{k} y_z^j\right] > \mathbb{E}\left[\sum_{j=1}^{k} y_r^j\right].$$

We note that the points generated by $O^2$ are more likely to belong to the minority class than other synthetic points, under $g_{\boldsymbol{w}}$, because they were generated based on $g_{\boldsymbol{w}}$. In addition, from Theorem 2 we observe that if $g_{\boldsymbol{w}}$ has lower misclassification rate at the points generated by $O^2$, then these points are more likely to belong to the minority class under the true distribution of the data. As a result, $O^2$ could outperform current randomization-based approaches, which is further validated in the numerical experiments. Again, the proof is included in the Appendix.

In Fig. 1, we illustrate the benefit of $O^2$ over SMOTE in a small two-dimensional example. We generate linearly separable data in $\mathbb{R}^2$. We then generate synthetic points with both $O^2$ and SMOTE and plot them along with those already in the training set and some "unobserved" minority class points that we did not use in training. From Figure 1, we observe that points generated by $O^2$ are close to the "unobserved" ones, while not being restricted in neighborhoods of those in the training set. On the other hand, points generated by SMOTE usually lie in neighborhoods of those in the training set and can deviate from "unobserved" minority class points. Moreover, we notice that, unlike SMOTE, $O^2$ can generate points in regimes where there exist few such points in the training set but are likely to include more minority class points. Basically, a decent base classifier, with some generalization power, enables $O^2$ to generate points across the true distribution, while SMOTE-based methods are restricted to existing points. Since Problem (3) is model dependent, we next derive an analytical formulation for two parametric models, LR, SVM and one non-parametric model, CART. For clarity, we denote our method as $O^2(A)$ where $A$ is one of the aforementioned models.

## 2.2 LOGISTIC REGRESSION

Assuming an input vector $\boldsymbol{z}$ as well as a LR model with learned coefficients $\boldsymbol{w}$, we want to maximize the probability of obtaining $y = 1$ over $\boldsymbol{z}$. In this case, the objective is as follows:

$$\boldsymbol{g_w}(\boldsymbol{z}) = \mathbb{P}(y = 1|\boldsymbol{z}) = \frac{\exp(\boldsymbol{\beta}^T \boldsymbol{z})}{1 + \exp(\boldsymbol{\beta}^T \boldsymbol{z})}. \tag{4}$$

In practice, we first train a LR model on the original dataset and learn the weights $\boldsymbol{w}$. Then, in order to generate synthetic minority class points we solve Problem (3) iteratively, with the function $g_{\boldsymbol{w}}(\boldsymbol{z})$ as defined in (4).

## 2.3 Support Vector Machines

SVMs have been proposed in Boser et al. (1992); Cortes & Vapnik (1995) and simply consist of learning an optimal hyperplane to separate points belonging to different classes. Assuming an input vector $z$ as well as an SVM model with learned coefficients $w$, we want our generated points to be on the side of the hyperplane that corresponds to minority class points. In this case, the objective is as follows

$$g_{w}(z) = \mathbb{P}(y = 1|z) = \text{sgn}(w^T z). \qquad (5)$$

As with LR, here we also start by training an SVM model and learning the weights $w$. Afterwards, we generate synthetic minority class points by solving Problem (3) iteratively, with the function $g_{w}(z)$ as defined in (5).

## 2.4 CART

The proposed framework also applies to non-parametric models. One broad category of such models is CART Breiman et al. (2017). CART feature splits of the form $a_j^T x \leq b_j$ such that for every input $x$, we reach a specific leaf by following the splits and classify $x$ based on the class of the majority of the points in that leaf. In our case, after fitting a CART, we locate leaves where the predicted class is the minority with high probability and keep track of the corresponding splits. Then, we generate synthetic minority class points by solving problem (3) with only the penalty terms in the objective and the tree splits as linear constraints. The resulting problem in this case is as follows:

$$\max_{z} \quad \lambda_1 \sum_{i \in [k-1]} \|z - z_i\|^2 + \lambda_2 \sum_{i \in [m]} \|z - x_i\|^2 - \lambda_3 \|z\|^2$$
$$\text{s.t.} \quad a_j^T z \leq b_j, \quad \forall j \in \mathcal{J}, \qquad (6)$$

where the set $\mathcal{J}$ consists of splits that lead to a region that is very likely to contain minority class points. Other tree models, i.e. Optimal Classification Trees (OCTs) Bertsimas & Dunn (2017; 2019) can also be used within our framework and may be desirable since they feature richer splits (e.g. OCTs with hyperplane splits).

## 2.5 Extra Filtering

An additional part of $O^2$ consists of an optional filtering step using a binary classifier as a way of filtering the candidate points. We propose training a classifier on the original dataset and then using it to filter out synthetic points that are not very likely to belong to the minority class. The choice of the classifier is up to the user and can be decided on a case-by-case basis. For our experiments, we use one of the current state-of-the-art models for tabular data, the TabNet classifier Arik & Pfister (2021). The optimization problems can be thought of as a generator for candidate points and the classifier as a way to further filter them. Therefore, both by construction and by filtering, $O^2$ points are optimized for belonging to the minority class.

We next present $O^2$ in pseudocode in Algorithm 1. The inputs of the algorithm are $\lambda_1, \lambda_2, \lambda_3$ the three penalty coefficients, $k$ the number of synthetic points to be generated by Problem (3), mdl the indicator specifying which model (LR, SVM or CART) we are going to use in the optimization problem, clf which indicates the classifier used in the filtering step and $X$ the original dataset. The output is $Z$, the oversampled version of $X$.

## 2.6 Extensions

Most real-world datasets contain both numerical and categorical features, while $O^2$ treats categorical (binary) features as numerical, i.e. as variables that take values in $[0, 1]$. To enhance performance, we consider using a multi-target classifier to generate the categorical part of the synthetic points by leveraging the numerical part. More precisely, we train a multi-target classifier on the numerical features, to predict the categorical ones. Afterwards, we create new synthetic points with numerical features only and then impute the corresponding categorical features from the prediction of the multi-target classifier. The procedure is summarized in pseudocode in Algorithm 2, see Appendix A.2. Another possible extension is related to datasets that feature targets with multiple classes. One approach for multi-class datasets is to turn them into multi-target datasets and use our method for

---

**Algorithm 1** $O^2$: Optimized Oversampling

---

**Input**: Data $\{\boldsymbol{x}_i, y_i\}_{i=1}^n$, initialization $\boldsymbol{z}^0$.
**Parameters**:

- $k$: number of points to generate by $O^2$
- $\lambda_1, \lambda_2, \lambda_3,$ : penalty coefficient for proximity to generated points, proximity to minority points, and for $\ell_2$ norm of the generated points
- mdl: classifier ($g_{\boldsymbol{w}}$) used in Problem (3) (default: LR)
- clf: classifier used for filtering (default: TabNet)

**Output**: Oversampled dataset $\boldsymbol{Z}$
 1: Train clf on $\boldsymbol{X}$
 2: Initialize $\boldsymbol{Z} = \boldsymbol{X}$, $\boldsymbol{V}$ as empty array, $i = 0$
 3: **while** $i < k$ **do**
 4:    Solve Problem (3) with input (mdl, $\lambda_1, \lambda_2, \lambda_3, \boldsymbol{V}$) and obtain $\boldsymbol{z}^*$.
 5:    **if** clf($\boldsymbol{z}^*$) $= 1$ **then**
 6:       Append $\boldsymbol{z}^*$ to $\boldsymbol{Z}$ and $\boldsymbol{V}$
 7:    **end if**
 8:    $i = i + 1$
 9: **end while**
10: **return** $\boldsymbol{Z}$

---

each target. In addition, we can adjust the optimization problem to minimize the distance of the classified point from a specific class, depending on the classifier we use.

## 3 NUMERICAL EXPERIMENTS

In this section, we test our method on 16 publicly available datasets. We use $O^2$ to generate minority class points using a model $A \in \{\text{LR, SVM, CART}\}$, denoted as $O^2(A)$. Then, we use a model $B \in \{\text{CART, LR}\}$ for the downstream classification task, denoted as $O^2(A,B)$. We evaluate all approaches in Python in terms of AUC and F1 score on a held out test set.

### 3.1 DATASET SELECTION

We experiment on a wide range of datasets, resembling those that arise in practice. The main factors in consideration when choosing the datasets are the following:

1. The number of rows ($n$). We mostly use datasets with rows in the thousands, which is common for tabular data, while also including a dataset with $n < 1,000$ (Indian Liver) as well as one with $n > 100,000$ (Skin).

2. The number of features ($p$). We mostly use datasets with number of features in the tens, while also including some datasets with $p \geq 100$ (AdaPrior, UsCrime, Scene).

3. The imbalance ratio (IR), defined as the number of minority class points over all the points in the dataset. We focus mainly on datasets with IR $\leq 0.1$.

4. The number of numerical and categorical features. We include datasets with only numerical attributes (Wine, Avila, etc.) and with both numerical and categorical features (AdaPrior, Churn, etc.).

A summary of the utilized datasets can be found in Table 6, in the Appendix. In multiclass classification problems, we transform the problem into an imbalanced binary classification problem using a one-vs-all encoding for some chosen class.

### 3.2 BENCHMARKING

More than one hundred oversampling methods are present in the literature so it is not practical to use all of them for benchmarking Kovács (2019). Similarly, there are numerous imbalanced datasets and

classifiers that can be used for experimentation. Our goal is not to present an in-depth comparison of all oversampling methods, but rather show that our method holds well against current state-of-the-art oversampling methods for a reasonable number of classifiers and datasets.

We select methods for benchmarking by considering the following factors: performance in the respective publications, popularity and date of publication. Out of all the proposed oversampling algorithms, we benchmark $O^2$ with SMOTE Chawla et al. (2002) (Sm), two of the most widely used SMOTE variants, Borderline1/2 SMOTE Han et al. (2005) (Bsm1, Bsm2), a method for oversampling that exhibits strong performance in practice, GSMOTE Douzas & Bacao (2019) (GSm) and one of the most recently proposed methods for oversampling, LoRAS Bej et al. (2021). We also include a baseline model (Bs) without oversampling. In terms of ML models, we use CART and LR.

For each dataset, we randomly split the data 70/30 into training and test sets, while respecting the original imbalance ratio. We oversample the training set and train all models on the original training set or the oversampled training set and report the AUC and F1 score on the held out test set. To reduce any randomness in the experiments we generate 10 copies of oversampled datasets for each method and report the average scores of all the runs on the test set. Finally, we also vary the number of generated points for each method for 5-10 different values and report the best results, in terms of AUC and F1 score, for each oversampling method on the test set.

We fix the penalty coefficients in our method to $\lambda_1 = 0.1, \lambda_2 = 0.05, \lambda_3 = 0.1$ for all datasets. These were tuned initially on a held out part of of the Wine dataset and gave good results for the other datasets, so there was no need to tune them again for the other datasets. Finally, we initialize the L-BFGS solver with the minority class points. The results for AUC and F1 score on the test set are summarized in Tables 1, 2 for CART as model B and in Tables 3, 4 for LR as model B.

Table 1: Comparison of test set AUC for CART.

| Dataset | Bs | Sm | GSm | BSm1 | BSm2 | LoRAS | $O^2$(LR) | $O^2$(SVM) | $O^2$(CART) |
|---|---|---|---|---|---|---|---|---|---|
| Wine | 65.5 | 68.9 | 68.8 | 68.2 | 66.3 | 67.9 | 69.2 | 68.2 | **69.6** |
| Avila | 82.3 | 81.2 | 80.2 | 83.4 | 82.4 | 84.3 | **87.1** | 85.5 | 84.2 |
| Mammography | 84.4 | 82.0 | 82.7 | 78.6 | 84.4 | 85.8 | 86.8 | 85.6 | **88.0** |
| Phoneme | 86.7 | 85.4 | 85.5 | 85 | 84.7 | **88.8** | 85.1 | 83.7 | 85.3 |
| AdaPrior | 86.8 | 87.0 | 87.3 | 87.2 | 87.0 | 86.9 | **87.4** | 87.2 | **87.4** |
| Churn | 85.3 | 86.2 | 85.6 | 86.4 | 86.0 | **88.8** | 86.6 | 86.5 | 88.1 |
| UsCrime | 85.7 | 85.7 | 85.9 | 82.5 | 81.3 | 87.0 | **87.2** | **87.2** | 85.9 |
| Letter Image | 96.0 | 96.1 | 95.6 | **97.0** | 90.5 | 96.6 | 96.2 | 96.1 | 96.1 |
| Pen Digits | 97.0 | 97.1 | 97.3 | 97.2 | 97.2 | 96.6 | **97.5** | 97.0 | 97.1 |
| Optical Digits | 92.4 | 91.7 | 91.5 | 91.7 | 91.2 | 83.3 | 91.9 | 91.8 | **92.8** |
| Finance | 60.9 | 71.3 | 67.4 | 66.7 | 70.9 | 52.8 | **83.1** | **83.1** | 82.6 |
| Satimage | 97.3 | 97.3 | 97.7 | 97.7 | 96.5 | **98.6** | 97.1 | 96.9 | 96.5 |
| Skin | **99.9** | **99.9** | **99.9** | 99.8 | 99.8 | 99.8 | **99.9** | **99.9** | **99.9** |
| IndianLiver | **71.8** | 68.6 | 64.6 | 63.5 | 64.2 | 63.1 | 65.3 | 69.1 | 68.6 |
| GermanCredit | 73.3 | 71.5 | 73.1 | 71.5 | 71.4 | **75.1** | 73.3 | 73.3 | 73.3 |
| Mozilla4 | 77.8 | 78.2 | **79.8** | 76.4 | 79.0 | 79.6 | 78.8 | 79.2 | 78.5 |
| **Rank** | 5.4 | 5.1 | 4.8 | 5.8 | 6.8 | 4.7 | **2.7** | 3.8 | 3.2 |

As Tables 1, 2 illustrate, when the classification task involves CART, $O^2$ has an edge over the other methods in terms of both AUC and F1 score, which becomes more pronounced in highly imbalanced datasets, e.g. UsCrime. In case the classification task involves LR, $O^2$ still has an edge over the other methods, although smaller, as illustrated in Tables 3, 4. Moreover, we notice that in this case, oversampling the minority class does not seem to make a significant difference in terms of out of sample AUC, a phenomenon common for all oversampling methods.

## 4 CASE STUDY: SPLENECTOMY

Interpretability and transparency are necessary in medicine, both for clinicians and for patients, so many works have introduced interpretable prognostic or prescriptive models for different medical applications Bertsimas et al. (2023). In this section, we empirically show that oversampling can increase both the performance and the interpretability of ML models. Specifically, we use CART and $O^2$ to predict the risk of mortality of patients with blunt spleen trauma.

Table 2: Comparison of test set F1 score for CART.

| Dataset | Bs | Sm | GSm | BSm1 | BSm2 | LoRAS | $O^2$(LR) | $O^2$(SVM) | $O^2$(CART) |
|---|---|---|---|---|---|---|---|---|---|
| Wine | 44.6 | **53.0** | **53.0** | 52.1 | 50.0 | 51.8 | 52.5 | 49.2 | 51.3 |
| Avila | 67.0 | 65.3 | 63.7 | 63.7 | 67.9 | 54.4 | **74.1** | 72.1 | 69.3 |
| Mammography | 57.9 | 63.1 | 61.6 | 63.2 | 62.6 | **65.6** | 58.0 | 58.0 | 58.3 |
| Phoneme | 77.8 | 77.2 | 77.4 | 77.2 | 77.5 | **78.7** | 77.1 | 76.7 | 77.0 |
| AdaPrior | 61.8 | 63.1 | **65.4** | 63.7 | 63.8 | 63.0 | 62.0 | 62.0 | 62.0 |
| Churn | 78.5 | 79.7 | 78.6 | 79.3 | 77.4 | **80.9** | 79.3 | 79.7 | 79.8 |
| UsCrime | 22.2 | 37.6 | 39.0 | 40.2 | 37.9 | **46.3** | 39.0 | 39.1 | 39.8 |
| Letter Image | 92.7 | 88.7 | 80.0 | 85.4 | 81.0 | 85.5 | **92.9** | 92.7 | **92.9** |
| Pen Digits | 95.6 | 95.6 | 95.6 | 95.5 | 94.7 | 90.5 | **95.7** | **95.7** | **95.7** |
| Optical Digits | **86.0** | 80.4 | 81.4 | 81.0 | 76.0 | 66.2 | 84.9 | 83.6 | 85.3 |
| Finance | 0.0 | **30.5** | 28.4 | 26.2 | 23.9 | 0.1 | 26.5 | 26.5 | 28.9 |
| Satimage | 95.2 | 95.6 | 95.8 | 94.7 | 92.9 | **96.4** | 95.8 | 95.3 | 95.3 |
| Skin | 99.8 | 99.8 | 99.8 | 99.8 | 99.6 | **99.9** | **99.9** | **99.9** | **99.9** |
| IndianLiver | 0.0 | 46.8 | 48.6 | 46.6 | 47.8 | **53.7** | 50.3 | 51.1 | 46.9 |
| GermanCredit | 46.3 | 49.5 | 50.0 | 48.9 | **50.7** | 46.2 | 46.3 | 46.3 | 46.3 |
| Mozilla4 | 11.1 | 26.9 | 22.3 | 24.1 | 15.4 | **28.8** | 11.7 | 11.5 | 11.3 |
| **Rank** | 6.4 | 4.3 | 4.3 | 5.1 | 6.0 | 4.3 | **3.8** | 4.4 | 4.1 |

Table 3: Comparison of test set AUC for LR.

| Dataset | Bs | Sm | GSm | BSm1 | BSm2 | LoRAS | $O^2$(LR) | $O^2$(SVM) | $O^2$(CART) |
|---|---|---|---|---|---|---|---|---|---|
| Wine | 73.3 | 73.3 | 73.6 | **74.1** | 73.9 | 73.0 | 73.4 | 73.3 | 73.3 |
| Avila | 73.7 | 73.7 | 73.7 | **75.0** | 74.7 | 74.9 | 73.9 | 74.0 | 73.9 |
| Mammography | 93.7 | 94.4 | 94.5 | **95.3** | 94.8 | 94.2 | 93.9 | 93.7 | 93.7 |
| Phoneme | **82.0** | **82.0** | **82.0** | 81.9 | 81.9 | 81.7 | 82.0 | 81.9 | 81.9 |
| AdaPrior | **90.0** | **90.0** | **90.0** | 89.9 | 89.9 | 89.9 | **90.0** | **90.0** | **90.0** |
| Churn | 78.1 | 78.8 | 78.8 | 78.9 | 79.0 | **79.4** | 78.1 | 77.9 | 78.1 |
| UsCrime | 92.1 | 91.9 | 92.1 | 91.9 | 92.1 | 91.8 | **92.2** | **92.2** | 90.9 |
| LetterImg | 99.0 | 99.0 | **99.1** | 99.0 | 99.0 | 99.0 | **99.1** | 99.0 | 99.0 |
| PenDigits | 98.0 | **98.2** | 98.1 | 98.1 | 98.1 | 98.1 | 98.1 | 98.0 | **98.2** |
| OpticalDigits | 97.6 | 97.7 | 97.7 | 97.4 | 97.3 | 97.6 | **97.8** | 97.6 | 97.6 |
| Finance | **92.5** | 92.1 | 92.1 | 92.4 | 92.2 | 91.8 | **92.5** | **92.5** | **92.5** |
| Satimage | 99.6 | 99.6 | 99.6 | 99.6 | 99.6 | **99.7** | 99.6 | 99.6 | 99.6 |
| Skin | **95.0** | **95.0** | **95.0** | 94.9 | 94.9 | 94.8 | **95.0** | **95.0** | **95.0** |
| IndianLiver | 81.3 | 81.7 | 81.8 | 81.5 | 81.2 | 82.0 | 81.5 | **83.0** | 81.6 |
| GermanCredit | 81.6 | 81.6 | 81.6 | 81.6 | **81.7** | 81.6 | 81.6 | **81.7** | 81.6 |
| Mozilla4 | 88.4 | 88.4 | 88.4 | **88.5** | **88.5** | 88.1 | 88.4 | 88.4 | 88.4 |
| **Rank** | 3.9 | 3.4 | 2.9 | 3.9 | 4.1 | 5.3 | **2.8** | 3.5 | 3.8 |

The spleen is an immunologic intra-abdominal organ on the left side of the body. Spleen injuries are common trauma-related injuries but there are few guidelines on selecting the optimal treatment for each patient Coccolini et al. (2017). The usual treatments are just observation and monitoring or splenectomy, which is the removal of the spleen Velmahos et al. (2010); Coccolini et al. (2017). We have historical records of patients with blunt spleen trauma who either underwent observation or splenectomy after admission to the Emergency Department (ED). The outcome is 1 in case of patient mortality and 0 otherwise. We use the following features, because they are available immediately after a patient enters the ED. We use the systolic blood pressure (sbp), the heart rate (pulse rate), the respiratory rate, the blood oxygen saturation (pulse oximetry), the body mass index (bmi), the total Glasgow Coma Scale score (totalgcs) and the age of the patient. The Glasgow Coma Scale (GCS) provides a score of the neurological response of a patient. It is measured by assigning scores to a patient's eye response, verbal response and motor response Hong et al. (2013). The total score, ranging from 3 to 15, is derived by adding these individual scores. A score of 3-8 can indicate a severe traumatic brain injury, while a score of 9-12 can indicate mild traumatic brain injury. Patients with normal neurological response generally have a score of 15. The normal values of these features are provided in Table 5.

There are 35,954 patients from different medical centers in the US. The mortality rate is 6 %, so the data are highly imbalanced. In this case, we retain only the patients with sbp less than 90. These patients are "hemodynamicaly unstable" and are usually recommended splenectomy Coccolini et al. (2017). Out of these patients, we seek to identify those with high risk of mortality, so that we can

Table 4: Comparison of test set F1 score for LR.

| Dataset | Bs | Sm | GSm | BSm1 | BSm2 | LoRAS | $O^2$(LR) | $O^2$(SVM) | $O^2$(CART) |
|---|---|---|---|---|---|---|---|---|---|
| Wine | 30.6 | 51.5 | 51.7 | **52.0** | **52.0** | 51.4 | 43.6 | 35.4 | 35.5 |
| Avila | 0.0 | 33.8 | 33.6 | **35.4** | 34.7 | 35.2 | 0.0 | 0.0 | 0.0 |
| Mammography | 44.8 | 55.2 | 55.9 | 59.7 | 56.3 | **62.0** | 56.9 | 56.4 | 56.4 |
| Phoneme | 53.7 | 61.0 | 61.0 | 60.7 | 60.8 | **63.9** | 62.1 | 61.7 | 61.8 |
| AdaPrior | 65.4 | 68.7 | 68.6 | 68.9 | 69.1 | **70.2** | 65.8 | 65.4 | 65.8 |
| Churn | 27.0 | 42.0 | 42.0 | 42.1 | 42.1 | **44.0** | 42.2 | 33.7 | 32.1 |
| UsCrime | 54.2 | 56.9 | 56.4 | 57.5 | 55.4 | 59.0 | **63.1** | 62.4 | 60.9 |
| LetterImg | 75.4 | 77.2 | 77.1 | 75.2 | 75.6 | 76.2 | 77.7 | **78.5** | **78.5** |
| PenDigits | 80.2 | 80.8 | 80.4 | 78.7 | 79.0 | 80.5 | 81.0 | 80.5 | **81.3** |
| OpticalDigits | 84.7 | 84.5 | 84.6 | 84.4 | 84.0 | 84.1 | **85.2** | 85.0 | 84.8 |
| Finance | 0.1 | **43.4** | 42.8 | 0.4 | 0.4 | 0.3 | 0.3 | 0.3 | 0.3 |
| Satimage | 94.5 | 95.0 | 95.0 | 93.5 | 93.7 | **95.2** | 95.1 | 94.6 | 94.8 |
| Skin | 80.7 | 84.5 | 84.5 | 81.7 | 82.5 | **86.1** | 82.5 | 82.3 | 82.5 |
| IndianLiver | 45.6 | 62.1 | 61.6 | 61.7 | 61.7 | 60.2 | **63.1** | 62.5 | 62.2 |
| GermanCredit | 63.0 | 64.3 | 64.4 | **65.0** | 64.6 | 61.4 | 64.6 | 64.3 | 64.5 |
| Mozilla4 | 76.8 | 76.6 | 76.7 | 76.6 | 76.6 | 70.4 | 76.8 | 76.8 | **76.9** |
| **Rank** | 7.6 | 4.4 | 4.8 | 5.1 | 5.1 | 4.3 | **3.0** | 4.6 | 4.0 |

Table 5: Normal Ranges for Vital Signs, GCS, and BMI Categories.

| Measurement | Normal Range/Categories |
|---|---|
| Heart Rate | 60–100 beats per minute (bpm) |
| Blood Pressure (Systolic) | 90–120 mmHg |
| Respiratory Rate | 12–20 breaths per minute (bpm) |
| Oxygen Saturation | 95–100% |
| Glasgow Coma Scale (GCS) | 15 (maximum score) |
| BMI | |
|    Underweight | less than 18.5 |
|    Normal Weight | 18.5–24.9 |
|    Overweight | 25–29.9 |
|    Obese | 30 or greater |

recommend splenectomy to the high-risk group and assign observation to the patients with lower risk. We used only the patients with sbp less than 90 who did not receive splenectomy in the data, since treatment affects the mortality outcome, which includes 1,220 patients with 16 % mortality rate.

The baseline model without oversampling has an AUC of 0.77 and an accuracy of 0.21 on the positive class, which is defined as $\frac{TP}{TP+FN}$, where TP are the true positives and FN are the false negatives. If we use $O^2$ with 30 synthetic minority class points, the AUC is 0.79 and the accuracy on the positive class is 0.45, which is a significant increase. Note that we are interested in the positive class accuracy, since we prioritize the prediction of mortality in the data.

While the baseline CART model has some intuitive splits, see Appendix A.9, e.g. predicts mortality for patients with low pulse rate or low oxygen saturation, the positive class accuracy is low. The oversampled CART model improves both the positive class accuracy and the AUC, while also containing intuitive splits. More specifically, from Fig. 2 we observe that there is a split on bmi less than 27.5, which is very close to the cutoff of 25 for overweight and obese patients. Similarly, the tree does not predict mortality for patients with very high oxygen saturation or relatively high blood pressure of 79.5. Of note, the threshold of 79.5 is close to 90, which is sometimes an arbitrary cutoff. Finally, the tree splits on totalgcs less than 10.5, which is more informative than the split of the first tree on 4.5. This is because totalgcs less than 4.5 is the same as totalgcs 3 or 4, which is very low, so the patient may not benefit from splenectomy, since they have severe neurological damage.

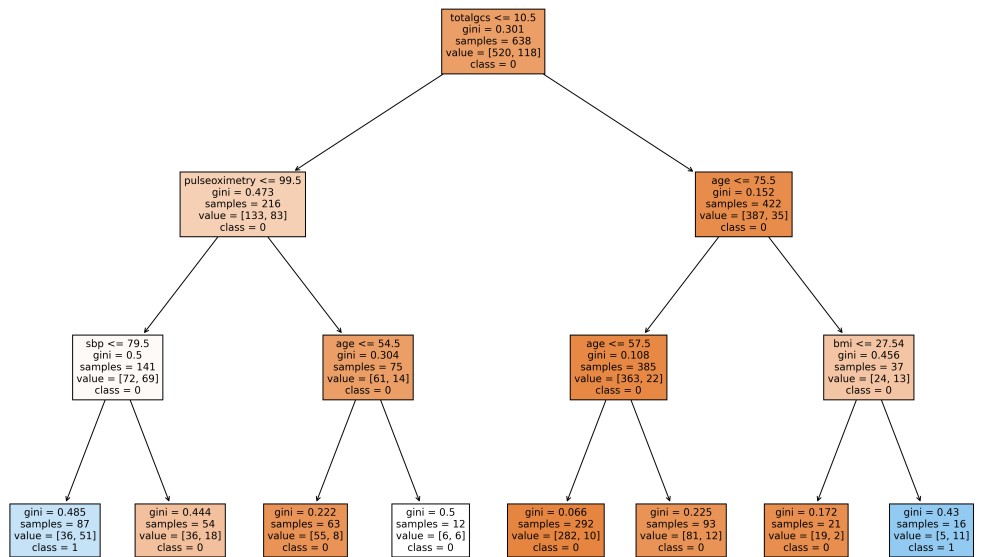

Figure 2: CART with oversampling on the spleen trauma dataset.

## 5 CONCLUSION

We have introduced a new oversampling framework, $O^2$, which in contrast to most oversampling methods, follows an optimization approach, leveraging the learning power of binary classification models. To the best of our knowledge, $O^2$ provides a deterministic alternative to the currently utilized methods, while it is easily amenable to modifications. In addition, $O^2$ can be used to improve synthetic points generated by other oversampling algorithms. In practice, it often outperforms the state-of-the-art methods. Moreover, it has a more pronounced edge when used on certain types of datasets, such as highly imbalanced - medium sized datasets, and the CART classifier. Finally, we demonstrated the value of $O^2$ on a real-world medical example, involving the prediction of mortality of patients with spleen trauma.

**Reproducibility** The approach can be reproduced through the code provided in the supplementary material. The code is based on Section 2 and Algorithm 1. The benchmarking procedure is described in 3 and in the Appendix.

**Ethics Statement** We acknoweldge the ICLR Code of Ethics. This work is free of potential conficts of interest, harmful insights and bias. The data for the splenectomy case study can only be provided upon request.

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

# A  APPENDIX

In Appendix A.1 we provide the proofs for Theorems 1 and 2. In Appendix A.2 we provide an extension of $O^2$ to datasets with both numerical and categorical features. In Appendix A.3 we provide details for the datasets that we used in the numerical experiments. In Appendix A.4 we provide details for the benchmarking procedure. In Appendix A.5 we show the benefit of initialzing $O^2$ from synthetic points obtained from other methods. In Appendix A.6 we demonstrate the value of the filtering step. In Appendix A.7 we have additional results for our method on datasets with both numerical and categorical attributes. In Appendix A.8 we compare the computational times for $O^2$ and the other oversampling methods. In Appendix A.9 we include additional results for the case study. In Appendix A.10 we perform an ablation study for the coefficients in the objective, and finally in Appendix A.11 we report $95\%$ confidence intervals for the results in the main paper.

## A.1  PROOFS

**Proof of Theorem 1**

*Proof.* It suffices to show that the objective function has Lipschitz continuous gradient Bertsekas (1997). Let $f(\boldsymbol{z})$ denote the objective function in problem (3), which has the following gradient

$$\nabla f(\boldsymbol{z}) \quad = \nabla g_{\boldsymbol{w}}(\boldsymbol{z}) + 2\lambda_1 \sum_{i \in [k-1]} (\boldsymbol{z} - \boldsymbol{z}_i) \\ + 2\lambda_2 \sum_{i \in [m]} (\boldsymbol{z} - \boldsymbol{x}_i) - 2\lambda_3 \boldsymbol{z}.$$

By the theorem's assumption,

$$\|\nabla g_{\boldsymbol{w}}(\boldsymbol{x}) - \nabla g_{\boldsymbol{w}}(\boldsymbol{y})\| \leq L\|\boldsymbol{x} - \boldsymbol{y}\|.$$

Thus, we have the following

$$\|\nabla f(\boldsymbol{x}) - \nabla f(\boldsymbol{y})\| = \|\nabla g_{\boldsymbol{w}}(\boldsymbol{x}) - \nabla g_{\boldsymbol{w}}(\boldsymbol{y}) \\ + 2\lambda_1 \sum_{i \in [k-1]} (\boldsymbol{x} - \boldsymbol{z}_i - \boldsymbol{y} + \boldsymbol{z}_i) \\ + 2\lambda_2 \sum_{i \in [m]} (\boldsymbol{x} - \boldsymbol{x}_i - \boldsymbol{y} + \boldsymbol{x}_i) \\ + 2\lambda_3(\boldsymbol{x} - \boldsymbol{y})\| \\ \leq \|\nabla g_{\boldsymbol{w}}(\boldsymbol{x}) - \nabla g_{\boldsymbol{w}}(\boldsymbol{y})\| \\ + 2\lambda_1 \sum_{i \in [k-1]} \|\boldsymbol{x} - \boldsymbol{y}\| \\ + 2\lambda_2 \sum_{i \in [m]} \|\boldsymbol{x} - \boldsymbol{y}\| + 2\lambda_3 \|\boldsymbol{x} - \boldsymbol{y}\| \\ \leq \tilde{L} \|\boldsymbol{x} - \boldsymbol{y}\|,$$

where $\tilde{L} = L + 2\lambda_1(k-1) + 2\lambda_2 m + 2\lambda_3$. Therefore, the objective has Lipschitz continuous gradient with constant $L_{obj} = \|\boldsymbol{w}\|^2 + 2\lambda_1(k-1) + 2\lambda_2 m + 2\lambda_3$.  □

**Proof of Theorem 2**

*Proof.* Consider a finite sample $(\boldsymbol{r}^1, \ldots, \boldsymbol{r}^k)$ obtained from an existing oversampling method and another finite sample $(\boldsymbol{z}^1, \ldots, \boldsymbol{z}^k)$ obtained from $O^2$ using $g_{\boldsymbol{w}}(\cdot)$. We assume that in both cases the points are distinct. Observe that each point $\boldsymbol{r}^i$ is either equal to a point $\boldsymbol{z}^j$ or it satisfies $g_{\boldsymbol{w}}(\boldsymbol{r}^i) \leq g_{\boldsymbol{w}}(\boldsymbol{z}^j)$ for all $j$, since $\boldsymbol{r}^i$ is feasible for Problem (2) for each $j$. We then have $g_{\boldsymbol{w}}(\boldsymbol{z}^j) \geq g_{\boldsymbol{w}}(\boldsymbol{r}^j)$. We round them to $\{-1, 1\}$ to obtain $\hat{z}^j \geq \hat{r}^j$. Assuming that the two samples are different by at least one point we then obtain that

$$\sum_{j=1}^{k} \hat{z}^j > \sum_{j=1}^{k} \hat{r}^j. \tag{7}$$

Let $y_z^j$ denote the true unknown label of the point $z^j$. Note that $y_z^j = \hat{z}^j(1-2\epsilon^j)$, where $\epsilon^j \in \{0,1\}$. Observe that either the model is right and $\epsilon^j = 0$ and $y_z^j = \hat{z}^j$ or the model is wrong in which case we have $\epsilon^j = 1$ and $y_z^j = -\hat{z}^j$. Then, $\epsilon^j$ is a random variable distributed as $\text{Ber}(1-p)$ for each $j$. So,

$$\mathbb{E}[y_z^j] = \mathbb{E}[\hat{z}^j(1-2\epsilon^j)] = \hat{z}^j - 2\hat{z}^j\mathbb{E}[\epsilon^j] = (2p-1)\hat{z}^j.$$

Similarly, we assume that $y_r^j$, the true unknown label of the point $r^j$, is defined as $y_r^j = \hat{r}^j(1-2\delta^j)$, where $\delta^j \sim \text{Ber}(1-\overline{p})$. We obtain

$$\mathbb{E}[y_r^j] = \mathbb{E}[\hat{r}^j(1-2\delta^j)] = \hat{r}^j - 2\hat{r}^j\mathbb{E}[\delta^j] = (2\overline{p}-1)\hat{r}^j.$$

and

$$\mathbb{E}\left[\sum_{j=1}^{k} y_z^j\right] = \sum_{j=1}^{k}\mathbb{E}\left[y_z^j\right] = (2p-1)\sum_{j=1}^{k}\hat{z}^j,$$

$$\mathbb{E}\left[\sum_{j=1}^{k} y_r^j\right] = \sum_{j=1}^{k}\mathbb{E}\left[y_r^j\right] = (2\overline{p}-1)\sum_{j=1}^{k}\hat{r}^j.$$

Moreover, since $p > 1/2$ from (7),

$$(2p-1)\sum_{j=1}^{k}\hat{z}^j > (2p-1)\sum_{j=1}^{k}\hat{r}^j \geq (2\overline{p}-1)\sum_{j=1}^{k}\hat{r}^j,$$

and the result follows. $\qquad\square$

## A.2 $O^2$ ALGORITHM FOR DATASETS WITH CATEGORICAL FEATURES

---

**Algorithm 2** $O^2$: Optimized Oversampling with numerical and categorical features

---

**Input**: Training dataset $X$, initialization $z^0$.
**Parameters**:

- $k, \lambda_1, \lambda_2, \lambda_3, \text{mdl}, \text{clf}$: same as in Algorithm 1
- multi-target: a multi-target classifier (default: TabNet)

**Output**: Oversampled dataset $Z$

 1: Train clf on $X$
 2: Split the data $X = [X_{\text{num}}, X_{\text{cat}}]$
 3: Train multi-target with $X_{\text{num}}$ as input and $X_{\text{cat}}$ as targets
 4: Initialize $Z = X$, $V$ as empty array, $i = 0$
 5: **while** $i < k$ **do**
 6:     Solve Problem (3) with input $(\text{mdl}, \lambda_1, \lambda_2, \lambda_3, V)$ and obtain $z_{\text{num}}$.
 7:     Obtain predictions $z_{\text{cat}} = \text{multi-target}(z_{\text{num}})$
 8:     Concatenate $z^* = [z_{\text{num}}, z_{\text{cat}}]$
 9:     **if** $\text{clf}(z^*) = 1$ **then**
10:         Append $z^*$ to $Z$ and $V$
11:     **end if**
12:     $i = i + 1$
13: **end while**
14: **return** $Z$

---

## A.3 DATASET DETAILS

**Datasets** A summary of the datasets can be found in Table 6. Eight of the utilized datasets come from the UCI Machine Learning Repository (UCI) Asuncion & Newman (2007)(Wine, Avila, Mammography, Phoneme, Satimage, Skin Segmentation, Indian Liver, German Credit), four come from

Table 6: Datasets used for benchmarking.

| Dataset | $n$ | $p$ | IR |
|---|---|---|---|
| Indian Liver | 583 | 9 | 0.29 |
| German Credit | 1000 | 19 | 0.30 |
| UsCrime | 1595 | 100 | 0.08 |
| Finance | 2570 | 82 | 0.04 |
| AdaPrior | 4562 | 14 | 0.24 |
| Wine | 3917 | 11 | 0.25 |
| Churn | 4250 | 20 | 0.14 |
| Phoneme | 4323 | 5 | 0.29 |
| Satimage | 4434 | 36 | 0.11 |
| Optical Digits | 4496 | 64 | 0.10 |
| Pen Digits | 8793 | 16 | 0.10 |
| Mammography | 8946 | 6 | 0.02 |
| Avila | 10430 | 10 | 0.10 |
| Letter Image | 14000 | 16 | 0.04 |
| Mozilla4 | 15545 | 5 | 0.33 |
| Skin Segmentation | 245056 | 3 | 0.21 |

the Python package imblearn.datasets Lemaître et al. (2017) (UsCrime, LetterImage, PenDigits, OpticalDigits), one from Kaggle (Finance) and three from the OpenML repository Vanschoren et al. (2014) (AdaPrior, Churn, Mozilla4). Although there exist multiple datasets with the desired characteristics, we decided to use the ones above as they are common in the oversampling literature Chawla et al. (2002); Han et al. (2005); Bej et al. (2021).

### A.4 BENCHMARKING DETAILS

For CART, we utilize a grid-search approach with 3-fold cross-validation on the training set to tune the following tree parameters: the loss metric (gini or entropy) and the tree's maximum depth (we use 7 possible depths from 3 to 100). To eliminate randomness in the results, we run the same grid-search 35 times in each training dataset and report the average of the results on the testing set. For LR, we average the results of 35 models with different "random seeds".

To be as fair as possible, we also vary the parameters of each oversampling method. The most important parameter of the SMOTE variants is the "ratio" which indicates the target ratio of minority to majority points in the final oversampled dataset. In our method, the most important parameter is "points" which, similarly to the "ratio", controls how many points we generate using our optimization algorithm. We do not tune these parameters but record the results of 5-10 different ratios, ranging from the smallest possible value to 1 (equal minority and majority points in the final dataset), and report the best one for each oversampling method, based on the AUC and F1 score on the test set.

To summarize, for each ratio and for each oversampling method, we generate 10 new datasets with that ratio and method, and we fit 35 models on each one using grid-search. We evaluate all 350 models on the testing set and use the average of the 350 test set AUCs and F1 scores as the performance metrics of the specific model, with the specific oversampling method and ratio. This procedure is also summarized in Algorithm 3. After collecting the test-set AUCs and F1 scores for the different ratios, we report the metrics corresponding to the best ratio for each method. We report results for the benchmarks and $O^2$ with either LR, SVM or CART as model A and CART or LR as model B.

### A.5 INITIALIZATION AND $O^2$ AS AN OPTIMIZER OF OTHER OVERSAMPLING METHODS

An important part in solving Problem (3), is the starting point $z^0$. Throughout the experiments so far we used the minority class points in the training set as warm starts. In this section, in Table 7 we examine whether we can do better by starting with points generated by LoRAS ($O^2$-LoRAS). For both cases we use $O^2$(LR, CART).

As Table 7 illustrates, five out of eight times the initialization from LoRAS results in better out of sample AUC. Moreover, there are examples where the difference is significant, i.e., the Avila dataset with an increase from 87.1 to 88.3 and the Phoneme dataset with an increase from 85.1 to 87.5.

---

**Algorithm 3** Training and evaluation steps

---

**Input**: Original dataset with train-test split $\boldsymbol{X}_{\text{train}}$, $\boldsymbol{X}_{\text{test}}$.
**Parameters**:

- $ovs_{mdl}$: oversampling method. Options: None, SMOTE, BorderlineSMOTE, G-SMOTE, LoRAS, $O^2$
- $k$: number of synthetic points to generate by $ovs_{mdl}$
- mdl: classification model. Options: CART, LR, RF

**Output**: Test set AUC and F1 for $(ovs_{mdl}, k, mdl)$
 1: Initialize res as an empty array, $i = 0$
 2: **while** $i < 10$ **do**
 3:    Initialize $\boldsymbol{Z}_{\text{train}}$ as an empty dataset
 4:    Use $ovs_{mdl}$ to generate $k$ new points using $\boldsymbol{X}_{\text{train}}$
 5:    $\boldsymbol{Z}_{\text{train}}$ is $\boldsymbol{X}_{\text{train}}$ and the newly generated points
 6:    $j = 0$
 7:    **while** $j < 35$ **do**
 8:      Fit $mdl$ on $\boldsymbol{Z}_{\text{train}}$ with a different random seed each time and the same grid-search parameters
 9:      Evaluate the trained $mdl$ on $\boldsymbol{X}_{\text{test}}$
10:      Add the AUC and F1 score to res
11:      $j = j + 1$
12:    **end while**
13:    $i = i + 1$
14: **end while**
15: **return**  Average of res for each metric

---

Table 7: Test set AUC comparison between $O^2$ and $O^2$-LoRAS.

| **Dataset** | $O^2$ | $O^2$**-LoRAS** |
|---|---|---|
| Wine | 69.2 | **70.0** |
| Avila | 87.1 | **88.3** |
| Mammography | **86.8** | 85.2 |
| Phoneme | 85.1 | **87.5** |
| AdaPrior | **87.4** | 86.9 |
| Churn | 86.6 | 86.6 |
| Pen Digits | **97.5** | 96.9 |
| Optical Digits | 91.9 | **92.6** |
| Satimage | 97.1 | **97.9** |

A drawback is that the suggested initialization increases computational time, since we need to run LoRAS first.

In general, the $O^2$ algorithm can be used to improve the probability of belonging to the minority class for points generated by other oversampling methods, by using them as warm starts for Problem (3). The intuitive explanation is that for a given point, the algorithm takes gradient steps in the direction that makes it more likely to belong to the minority class.

### A.6   THE EFFECT OF FILTERING

In this section, we illustrate the importance of the filtering step of $O^2$. To see how it affects performance, we apply $O^2$ with and without the filtering step and compare the average improvement in terms of the AUC on the test set over various numbers of synthetic points. The results are illustrated in Table 8. Columns w and w/o correspond to the filtering and no-filtering cases, while the p-value corresponds to the null hypothesis of equal average test set AUC among filtering and no-filtering. For each number of synthetic points, the results are averaged over 10 runs. Further, each average is taken over five different numbers of generated points.

Table 8: The effect of filtering on test set AUC.

| Dataset | $O^2$(LR,CART) | | | $O^2$(LR,LR) | | |
|---|---|---|---|---|---|---|
| | w/o | w | p-value | w/o | w | p-value |
| Wine | 67.3 | **68.5** | $10^{-4}$ | 73.3 | **73.4** | 0.08 |
| Mammography | 81.2 | **84.8** | 0.01 | 91.4 | **93.6** | 0.34 |
| Avila | 82.4 | **86.2** | 0.001 | 73.4 | **73.8** | 0.003 |
| Phoneme | 83.5 | 83.5 | 1 | 81.8 | **81.9** | 0.01 |
| Churn | 84.5 | **86.4** | 0.04 | **78.3** | 77.8 | 0.28 |
| Ada Prior | 85.5 | **86.8** | 0.11 | 89.9 | **90** | 0.01 |
| UsCrime | 67.5 | **84.2** | $10^{-4}$ | 91.6 | **92.3** | 0.001 |
| LetterImg | 95.6 | **95.8** | 0.27 | 99 | 99 | 1 |
| PenDigits | 96.9 | **97.1** | 0.28 | 97.9 | **98.1** | 0.01 |
| OpticalDigits | **90.3** | 89.9 | 0.66 | **97.6** | 97.5 | 0.15 |

As Table 8 illustrates, in most datasets the filtering step results in an improved AUC, where the difference is significant both in magnitude and statistically. The intuition behind this is that since we restrict our generator, i.e. Problem (3), to generate points that differ from previous ones, it might be the case that more attention is placed on the penalty terms compared to the probability term in the objective. In this case, the filtering step allows us to cut off candidate points that are not very likely to belong to the minority class according to another model.

## A.7 $O^2$ ON DATASETS WITH BOTH NUMERICAL AND CATEGORICAL ATTRIBUTES.

In this section, we demonstrate the value of Algorithm 2 on datasets with both numerical and categorical attributes. The experiments feature datasets where most attributes are numerical (Churn), as well as datasets where most attributes are categorical (German Credit). A comparison between Algorithms 1 and 2 in terms of AUC on the test set is provided in Table 9. The p-value corresponds to the null hypothesis of equal average AUC. The average is taken over five different numbers of generated points.

Table 9: Test set AUC for Algorithms 1 and 2.

| Dataset | Attributes | | $O^2$(LR,CART) | | | $O^2$(LR,LR) | | |
|---|---|---|---|---|---|---|---|---|
| | Num | Cat | Alg. 1 | Alg. 2 | p-value | Alg. 1 | Alg. 2 | p-value |
| German Credit | 7 | 12 | 69.7 | 72.8 | 0.03 | 81.6 | 81.5 | 0.15 |
| Mozilla4 | 4 | 1 | 78.2 | 78.1 | 0.62 | 88.4 | 88.4 | 0.95 |
| Indian Liver | 9 | 1 | 66.6 | 67.2 | 0.73 | 81.5 | 81.6 | $10^{-4}$ |
| Churn | 18 | 2 | 86.2 | 87.2 | 0.19 | 77.8 | 77.6 | 0.53 |
| AdaPrior | 6 | 8 | 87.2 | 87.1 | 0.46 | 90.0 | 89.8 | 0.42 |

As Table 9 illustrates, overall the use of the multi-target classifier improves the AUC, as long as there are sufficiently many numerical attributes with some predictive power on the categorical ones. There are also cases, such as the German Credit dataset, where the number of numerical attributes is much smaller than the categorical but since they have predictive power, using the multi-target classifier helps. In practice when the numerical attributes have predictive power on the categorical ones, the multi-target classifier can be beneficial.

## A.8 COMPUTATIONAL TIMES

An important question is how computational times compare between $O^2$ and the SMOTE variants. The latter are based on randomization and thus can generate synthetic points quite fast, while $O^2$ requires solving an optimization problem to generate a synthetic point. In Table 10, we compare the computational times of both approaches for generating 100 synthetic minority class points, averaged over 4 runs in each dataset. We observe that $O^2$ requires more computational time than SMOTE and its variants, since it involves solving multiple optimization problems. Of note, all methods were able to generate 100 synthetic minority class points in seconds. The SMOTE variants need less than a

second, while $O^2$ needs 26 seconds on average. So the computational cost is relatively low, because in practice we run the algorithm only once, taking a few seconds before the classification problem.

Table 10: Computational times in seconds for $O^2$ and SMOTE variants.

| Dataset | Sm | GSm | BSm1 | $O^2$ |
|---|---|---|---|---|
| Wine | 0.016 | 0.049 | 0.067 | 35.739 |
| Mammography | 0.002 | 0.024 | 0.017 | 5.355 |
| Avila | 0.014 | 0.087 | 0.111 | 38.325 |
| Phoneme | 0.005 | 0.016 | 0.012 | 23.204 |
| UsCrime | 0.003 | 0.028 | 0.025 | 27.425 |

### A.9 ADDITIONAL RESULTS FOR CASE STUDY

We provide the CART without oversampling.

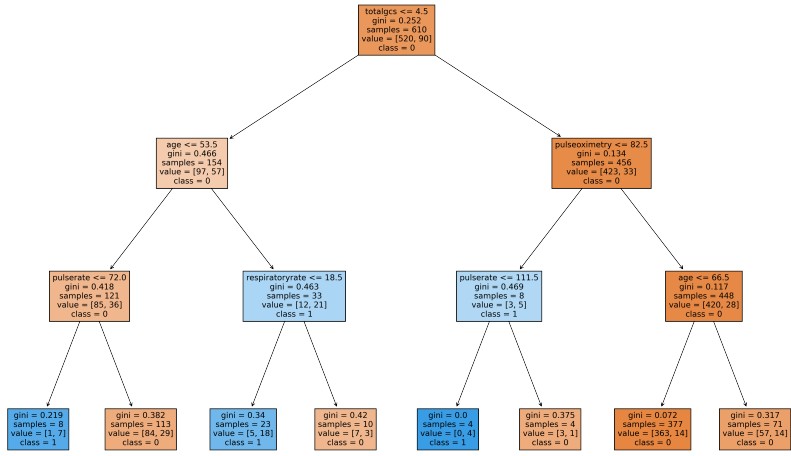

Figure 3: CART on the spleen trauma dataset.

### A.10 ABLATION STUDY FOR OBJECTIVE COEFFICIENTS

In this section, we investigate the effect of the different objective coefficients on the out of sample AUC to further support our final choice for $\lambda_1, \lambda_2$ and $\lambda_3$. The parameter $\lambda_3$ multiplies the term $\|x\|^2$ which is there in order to avoid unbounded solutions and therefore it does not need a lot of tuning. We next explore the effect of different values of the parameters $\lambda_1$ and $\lambda_2$ on the test set AUC of the Wine dataset. We generate 100 synthetic minority class points with $O^2$ by varying the values of $\lambda_1$ and $\lambda_2$ and evaluate the AUC on the fixed test set. We take $\lambda_1, \lambda_2 \in \{0.001, 0.01, 0.05, 0.1, 0.3, 0.5, 0.7, 1\}$. The results are illustrated in Figure 4.

From Figure 4, we observe that the higher AUC is achieved for values of $\lambda_1$ and $\lambda_2$ that are around 0.01. We see whenever one of them is around 1 the out of sample AUC decreases, which is reasonable since there is equal weight placed on the points being different than the current ones and on the minority class probability.

Figure 4: Out of sample AUC for different values of the objective coefficients on the test set for the Wine dataset.

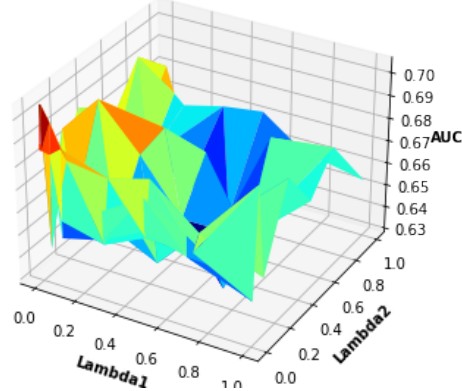

## A.11 STATISTICAL ANALYSIS OF THE RESULTS

We provide the 95 % confidence intervals corresponding to the experiments of Tables 1, 3, 2, and 4, in Tables 11, 12, 13, and 14, respectively. Each entry in the main tables is the result of 350 averaged entries, so our sample size is 350. We see that the intervals are small and do not overlap, which increases the confidence in our results.

Table 11: The 95 % confidence intervals for the AUC comparison for CART.

| Dataset | Bs | Sm | GSm | BSm1 | BSm2 | LoRAS | $O^2$(LR) | $O^2$(SVM) | $O^2$(CART) |
|---|---|---|---|---|---|---|---|---|---|
| Wine | (65.5, 65.5) | (68.9, 68.9) | (68.8, 68.8) | (68.2, 68.2) | (66.3, 66.3) | (67.9, 67.9) | (69.2, 69.2) | (68.2, 68.2) | (69.6, 69.6) |
| Avila | (82.3, 82.3) | (81.2, 81.2) | (80.19, 80.2) | (83.39, 83.41) | (82.4, 82.4) | (84.3, 84.3) | (87.1, 87.1) | (85.5, 85.5) | (84.2, 84.2) |
| Mammography | (84.39, 84.41) | (82.0, 82.0) | (82.7, 82.7) | (78.6, 78.6) | (84.39, 84.41) | (85.8, 85.8) | (86.8, 86.8) | (85.6, 85.6) | (88.0, 88.0) |
| Phoneme | (86.7, 86.7) | (85.4, 85.4) | (85.5, 85.5) | (85.0, 85.0) | (84.7, 84.7) | (88.8, 88.8) | (85.1, 85.1) | (83.7, 83.7) | (85.3, 85.3) |
| AdaPrior | (86.8, 86.8) | (87.0, 87.0) | (87.3, 87.3) | (87.2, 87.2) | (87.0, 87.0) | (86.9, 86.9) | (87.4, 87.4) | (87.2, 87.2) | (87.4, 87.4) |
| Churn | (85.3, 85.3) | (86.2, 86.2) | (85.6, 85.6) | (86.4, 86.4) | (86.0, 86.0) | (88.8, 88.8) | (86.6, 86.6) | (86.5, 86.5) | (88.1, 88.1) |
| UsCrime | (85.69, 85.71) | (85.69, 85.71) | (85.9, 85.9) | (82.49, 82.51) | (81.29, 81.31) | (87.0, 87.0) | (87.2, 87.2) | (87.2, 87.2) | (85.88, 85.92) |
| LetterImg | (96.0, 96.0) | (96.1, 96.1) | (95.6, 95.6) | (97.0, 97.0) | (90.49, 90.51) | (96.6, 96.6) | (96.2, 96.2) | (96.1, 96.1) | (96.1, 96.1) |
| PenDigits | (97.0, 97.0) | (97.1, 97.1) | (97.3, 97.3) | (97.2, 97.2) | (97.2, 97.2) | (96.59, 96.61) | (97.5, 97.5) | (97.0, 97.0) | (97.1, 97.1) |
| OpticalDigits | (92.4, 92.4) | (91.7, 91.7) | (91.5, 91.5) | (91.7, 91.7) | (91.2, 91.2) | (83.3, 83.3) | (91.9, 91.9) | (91.8, 91.8) | (92.8, 92.8) |
| Finance | (60.9, 60.91) | (71.29, 71.31) | (67.39, 67.41) | (66.69, 66.7) | (70.88, 70.92) | (52.8, 52.8) | (83.1, 83.1) | (83.1, 83.1) | (82.58, 82.62) |
| Satimage | (97.3, 97.3) | (97.3, 97.3) | (97.7, 97.7) | (97.7, 97.7) | (96.5, 96.5) | (98.6, 98.6) | (97.1, 97.1) | (96.9, 96.9) | (96.5, 96.5) |
| Skin | (99.9, 99.9) | (99.9, 99.9) | (99.9, 99.9) | (99.8, 99.8) | (99.8, 99.8) | (99.8, 99.8) | (99.9, 99.9) | (99.9, 99.9) | (99.9, 99.9) |
| IndianLiver | (71.79, 71.81) | (68.59, 68.61) | (64.6, 64.6) | (63.5, 63.5) | (64.19, 64.21) | (63.1, 63.1) | (65.3, 65.31) | (69.09, 69.11) | (68.59, 68.61) |
| GermanCredit | (73.29, 73.31) | (71.49, 71.51) | (73.09, 73.11) | (71.49, 71.51) | (71.39, 71.41) | (75.1, 75.1) | (73.3, 73.3) | (73.3, 73.3) | (73.3, 73.3) |
| Mozilla4 | (77.8, 77.8) | (78.2, 78.2) | (79.8, 79.8) | (76.4, 76.4) | (79.0, 79.0) | (79.6, 79.6) | (78.8, 78.8) | (79.2, 79.2) | (78.49, 78.51) |

Table 12: The 95 % confidence intervals for the AUC comparison for LR.

| Dataset | Bs | Sm | GSm | BSm1 | BSm2 | LoRAS | $O^2$(LR) | $O^2$(SVM) | $O^2$(CART) |
|---|---|---|---|---|---|---|---|---|---|
| Wine | (73.3, 73.3) | (73.3, 73.3) | (73.6, 73.6) | (74.1, 74.1) | (73.9, 73.9) | (73.0, 73.0) | (73.4, 73.4) | (73.3, 73.3) | (73.3, 73.3) |
| Avila | (73.7, 73.7) | (73.7, 73.7) | (73.7, 73.7) | (75.0, 75.0) | (74.7, 74.7) | (74.9, 74.9) | (73.9, 73.9) | (74.0, 74.0) | (73.9, 73.9) |
| Mammography | (93.7, 93.7) | (94.4, 94.4) | (94.5, 94.5) | (95.3, 95.3) | (94.8, 94.8) | (94.2, 94.2) | (93.9, 93.9) | (93.7, 93.7) | (93.7, 93.7) |
| Phoneme | (82.0, 82.0) | (82.0, 82.0) | (82.0, 82.0) | (81.9, 81.9) | (81.9, 81.9) | (81.7, 81.7) | (82.0, 82.0) | (81.9, 81.9) | (81.9, 81.9) |
| AdaPrior | (90.0, 90.0) | (90.0, 90.0) | (90.0, 90.0) | (89.9, 89.9) | (89.9, 89.9) | (89.9, 89.9) | (90.0, 90.0) | (90.0, 90.0) | (90.0, 90.0) |
| Churn | (78.1, 78.1) | (78.8, 78.8) | (78.8, 78.8) | (78.9, 78.9) | (79.0, 79.0) | (79.4, 79.4) | (78.1, 78.1) | (77.9, 77.9) | (78.1, 78.1) |
| UsCrime | (92.1, 92.1) | (91.9, 91.9) | (92.1, 92.1) | (91.9, 91.9) | (92.1, 92.1) | (91.8, 91.8) | (92.2, 92.2) | (92.2, 92.2) | (90.9, 90.9) |
| LetterImg | (99.0, 99.0) | (99.0, 99.0) | (99.1, 99.1) | (99.0, 99.0) | (99.0, 99.0) | (99.0, 99.0) | (99.1, 99.1) | (99.0, 99.0) | (99.0, 99.0) |
| PenDigits | (98.0, 98.0) | (98.2, 98.2) | (98.1, 98.1) | (98.1, 98.1) | (98.1, 98.1) | (98.1, 98.1) | (98.1, 98.1) | (98.0, 98.0) | (98.2, 98.2) |
| OpticalDigits | (97.53, 97.67) | (97.63, 97.77) | (97.63, 97.77) | (97.33, 97.47) | (97.23, 97.37) | (97.6, 97.6) | (97.8, 97.8) | (97.6, 97.6) | (97.6, 97.6) |
| Finance | (92.5, 92.5) | (92.1, 92.1) | (92.1, 92.1) | (92.4, 92.4) | (92.2, 92.2) | (91.8, 91.8) | (92.5, 92.5) | (92.5, 92.5) | (92.5, 92.5) |
| Satimage | (99.6, 99.6) | (99.6, 99.6) | (99.6, 99.6) | (99.6, 99.6) | (99.6, 99.6) | (99.7, 99.7) | (99.6, 99.6) | (99.6, 99.6) | (99.6, 99.6) |
| Skin | (95.0, 95.0) | (95.0, 95.0) | (95.0, 95.0) | (94.9, 94.9) | (94.9, 94.9) | (94.8, 94.8) | (95.0, 95.0) | (95.0, 95.0) | (95.0, 95.0) |
| IndianLiver | (81.3, 81.3) | (81.7, 81.7) | (81.8, 81.8) | (81.5, 81.5) | (81.2, 81.2) | (82.0, 82.0) | (81.5, 81.5) | (83.0, 83.0) | (81.6, 81.6) |
| GermanCredit | (81.6, 81.6) | (81.6, 81.6) | (81.6, 81.6) | (81.6, 81.6) | (81.7, 81.7) | (81.6, 81.6) | (81.6, 81.6) | (81.7, 81.7) | (81.6, 81.6) |
| Mozilla4 | (88.4, 88.4) | (88.4, 88.4) | (88.4, 88.4) | (88.5, 88.5) | (88.5, 88.5) | (88.1, 88.1) | (88.4, 88.4) | (88.4, 88.4) | (88.4, 88.4) |

Table 13: The 95 % confidence intervals for the F1 score comparison for CART.

| Dataset | Bs | Sm | GSm | BSm1 | BSm2 | LoRAS | $O^2$(LR) | $O^2$(SVM) | $O^2$(CART) |
|---|---|---|---|---|---|---|---|---|---|
| Wine | (44.6, 44.6) | (53.0, 53.0) | (53.0, 53.0) | (52.1, 52.1) | (50.0, 50.0) | (51.8, 51.8) | (52.5, 52.5) | (49.2, 49.2) | (51.3, 51.3) |
| Avila | (67.0, 67.0) | (65.3, 65.3) | (63.69, 63.71) | (63.7, 63.7) | (67.89, 67.91) | (54.4, 54.4) | (74.1, 74.1) | (72.1, 72.1) | (69.29, 69.31) |
| Mammography | (57.9, 57.91) | (63.1, 63.1) | (61.59, 61.6) | (63.2, 63.2) | (62.59, 62.61) | (65.6, 65.6) | (57.99, 58.01) | (57.99, 58.01) | (58.29, 58.31) |
| Phoneme | (77.8, 77.8) | (77.2, 77.2) | (77.4, 77.4) | (77.2, 77.2) | (77.5, 77.5) | (78.7, 78.7) | (77.1, 77.1) | (76.7, 76.7) | (77.0, 77.0) |
| AdaPrior | (61.8, 61.8) | (63.1, 63.1) | (65.4, 65.4) | (63.7, 63.7) | (63.8, 63.8) | (63.0, 63.0) | (62.0, 62.0) | (62.0, 62.0) | (62.0, 62.0) |
| Churn | (78.5, 78.5) | (79.7, 79.7) | (78.6, 78.6) | (79.3, 79.3) | (77.4, 77.4) | (80.9, 80.9) | (79.3, 79.3) | (79.7, 79.7) | (79.8, 79.8) |
| UsCrime | (22.2, 22.2) | (37.59, 37.61) | (38.99, 39.01) | (40.19, 40.21) | (37.89, 37.91) | (46.29, 46.31) | (38.99, 39.01) | (39.1, 39.1) | (39.79, 39.81) |
| LetterImg | (92.7, 92.7) | (88.7, 88.7) | (79.99, 80.01) | (85.39, 85.41) | (80.97, 81.03) | (85.49, 85.51) | (92.9, 92.9) | (92.7, 92.7) | (92.9, 92.9) |
| PenDigits | (95.6, 95.6) | (95.6, 95.6) | (95.6, 95.6) | (95.5, 95.5) | (94.69, 94.7) | (90.49, 90.51) | (95.7, 95.7) | (95.7, 95.7) | (95.7, 95.7) |
| OpticalDigits | (85.99, 86.01) | (80.39, 80.41) | (81.39, 81.41) | (80.99, 81.01) | (75.99, 76.01) | (66.15, 66.25) | (84.9, 84.9) | (83.6, 83.6) | (85.3, 85.3) |
| Finance | (-0.01, 0.01) | (30.49, 30.51) | (28.39, 28.41) | (26.19, 26.21) | (23.88, 23.92) | (0.1, 0.1) | (26.5, 26.5) | (26.5, 26.5) | (28.89, 28.91) |
| Satimage | (95.2, 95.2) | (95.6, 95.6) | (95.8, 95.8) | (94.7, 94.7) | (92.89, 92.91) | (96.4, 96.4) | (95.8, 95.8) | (95.3, 95.3) | (95.3, 95.3) |
| IndianLiver | (-0.04, 0.04) | (46.79, 46.81) | (48.59, 48.61) | (46.59, 46.61) | (47.79, 47.81) | (53.7, 53.7) | (50.3, 50.3) | (51.09, 51.11) | (46.89, 46.91) |
| GermanCredit | (46.29, 46.31) | (49.49, 49.51) | (49.99, 50.01) | (48.89, 48.91) | (50.69, 50.71) | (46.2, 46.2) | (46.29, 46.31) | (46.29, 46.31) | (46.3, 46.3) |
| Mozilla4 | (11.1, 11.1) | (26.89, 26.91) | (22.28, 22.32) | (24.09, 24.11) | (15.38, 15.42) | (28.8, 28.8) | (11.69, 11.71) | (11.48, 11.52) | (11.29, 11.31) |

Table 14: The 95 % confidence intervals for the F1 score comparison for LR.

| Dataset | Bs | Sm | GSm | BSm1 | BSm2 | LoRAS | $O^2$(LR) | $O^2$(SVM) | $O^2$(CART) |
|---|---|---|---|---|---|---|---|---|---|
| Wine | (30.6, 30.6) | (51.5, 51.5) | (51.7, 51.7) | (52.0, 52.0) | (52.0, 52.0) | (51.4, 51.4) | (43.6, 43.6) | (35.4, 35.4) | (35.5, 35.5) |
| Avila | (0.0, 0.0) | (33.8, 33.8) | (33.6, 33.6) | (35.4, 35.4) | (34.7, 34.7) | (35.2, 35.2) | (0.0, 0.0) | (0.0, 0.0) | (0.0, 0.0) |
| Mammography | (44.8, 44.8) | (55.2, 55.2) | (55.9, 55.9) | (59.7, 59.7) | (56.3, 56.3) | (62.0, 62.0) | (56.9, 56.9) | (56.4, 56.4) | (56.4, 56.4) |
| Phoneme | (53.7, 53.7) | (61.0, 61.0) | (61.0, 61.0) | (60.7, 60.7) | (60.8, 60.8) | (63.9, 63.9) | (62.1, 62.1) | (61.7, 61.7) | (61.8, 61.8) |
| AdaPrior | (65.4, 65.4) | (68.7, 68.7) | (68.6, 68.6) | (68.9, 68.9) | (69.1, 69.1) | (70.2, 70.2) | (65.8, 65.8) | (65.4, 65.4) | (65.8, 65.8) |
| Churn | (27.0, 27.0) | (42.0, 42.0) | (42.0, 42.0) | (42.1, 42.1) | (42.1, 42.1) | (44.0, 44.0) | (42.2, 42.2) | (33.7, 33.7) | (32.1, 32.1) |
| UsCrime | (54.19, 54.21) | (56.9, 56.9) | (56.4, 56.4) | (57.5, 57.5) | (55.39, 55.41) | (59.0, 59.0) | (63.1, 63.1) | (62.4, 62.4) | (60.9, 60.9) |
| LetterImg | (75.4, 75.4) | (77.2, 77.2) | (77.1, 77.1) | (75.2, 75.2) | (75.6, 75.6) | (76.2, 76.2) | (77.7, 77.7) | (78.5, 78.5) | (78.5, 78.5) |
| PenDigits | (80.2, 80.2) | (80.8, 80.8) | (80.4, 80.4) | (78.7, 78.7) | (79.0, 79.0) | (80.5, 80.5) | (81.0, 81.0) | (80.5, 80.5) | (81.3, 81.3) |
| OpticalDigits | (84.64, 84.76) | (84.44, 84.56) | (84.54, 84.66) | (84.34, 84.46) | (83.94, 84.06) | (84.1, 84.1) | (85.2, 85.2) | (85.0, 85.0) | (84.8, 84.8) |
| Finance | (0.1, 0.1) | (43.4, 43.4) | (42.8, 42.8) | (0.36, 0.36) | (0.41, 0.43) | (0.29, 0.29) | (0.32, 0.32) | (0.32, 0.32) | (0.32, 0.32) |
| Satimage | (94.5, 94.5) | (95.0, 95.0) | (95.0, 95.0) | (93.5, 93.5) | (93.7, 93.7) | (95.2, 95.2) | (95.1, 95.1) | (94.6, 94.6) | (94.8, 94.8) |
| IndianLiver | (45.6, 45.6) | (62.1, 62.1) | (61.6, 61.6) | (61.7, 61.7) | (61.7, 61.7) | (60.2, 60.2) | (63.1, 63.1) | (62.5, 62.5) | (62.2, 62.2) |
| GermanCredit | (63.0, 63.0) | (64.3, 64.3) | (64.4, 64.4) | (65.0, 65.0) | (64.6, 64.6) | (61.4, 61.4) | (64.6, 64.6) | (64.3, 64.3) | (64.5, 64.5) |
| Mozilla4 | (76.8, 76.8) | (76.6, 76.6) | (76.7, 76.7) | (76.6, 76.6) | (76.6, 76.6) | (70.4, 70.4) | (76.8, 76.8) | (76.8, 76.8) | (76.9, 76.9) |

