# OpenReview forum: "Optimized Oversampling"
_ICLR.cc/2025/Conference — Submitted to ICLR 2025_

### Official Review · Reviewer_Gemb · 2024-10-20

**Soundness:** 2
**Presentation:** 2
**Contribution:** 2
**Rating:** 5
**Confidence:** 4

**Summary:**

This paper introduces Optimized Oversampling ($O^2$), a new framework to address class imbalance in classification tasks. $O^2$ generates synthetic minority class points by maximizing their probability of belonging to the minority class, as estimated by a trained model. Theoretical analysis shows $O^2$ has an advantage over other methods, and it outperforms state-of-the-art techniques on 16 imbalanced datasets, especially with the CART classifier.$O^2$ offers a deterministic, optimization-based approach that enhances synthetic points from other methods, proving particularly effective on highly imbalanced, medium-sized datasets and in real-world applications.

This method approaches sample generation from the perspective of objective function optimization, but it may not be effective for the more current issues, such as image datasets with a long-tailed distribution. Furthermore, the method for generating new samples for binary imbalance problems is not novel.

**Strengths:**

1. This paper proposes a framework for generating high-probability minority class samples and provides a theoretical analysis of its effectiveness.

2. The proposed method is simple and effective in terms of both process and outcome, which is commendable.

3. The authors applied the proposed method to medical research, effectively predicting the mortality of patients with spleen trauma.

**Weaknesses:**

1. The research topic is outdated, as these issues were proposed 20 years ago. The algorithms being compared, except for LoRAS, are all fairly old.

2. The parameters likely influence the experimental results. The impact of parameters $\lambda_1$, $\lambda_2$, and $\lambda_3$ on the results needs further discussion.

3. The main issue with this paper is that it does not address the current problem of long-tailed image distributions in imbalanced datasets. For high-dimensional, multi-class image datasets such as CIFAR10-LT, CIFAR100-LT, and iNaturalist2018, the effectiveness of the proposed oversampling method remains to be validated.

If the authors can demonstrate that their method is effective on these long-tailed datasets, I would change my opinion of this paper from negative to positive.

**Questions:**

See Weaknesses.

---

### Official Review · Reviewer_jm3W · 2024-10-29

**Soundness:** 2
**Presentation:** 2
**Contribution:** 2
**Rating:** 5
**Confidence:** 4

**Summary:**

The paper introduces a novel oversampling framework called Optimized Oversampling (O2) aimed at addressing the issue of class imbalance in machine learning classification tasks. The authors propose a method that generates synthetic minority class points by maximizing the probability of belonging to the minority class, as estimated by a trained classification model. The paper presents theoretical guarantees that the points generated by O2 are more likely to belong to the minority class compared to those generated by existing methods. The authors benchmark O2 against state-of-the-art oversampling techniques on 16 publicly available imbalanced datasets using Classification Trees (CART) and Logistic Regression (LR) for downstream classification tasks. The results indicate that O2 outperforms existing methods, particularly in highly imbalanced datasets. Additionally, a case study is presented, demonstrating the application of O2 in predicting mortality risk in patients with blunt spleen trauma, highlighting improvements in both performance and interpretability.

**Strengths:**

1. The paper provides theoretical backing for the effectiveness of O2.
2. The authors benchmark O2 against a variety of state-of-the-art oversampling methods across multiple datasets.
3. The case study on predicting mortality risk in patients with blunt spleen trauma illustrates the practical implications of the proposed method.

**Weaknesses:**

1. While the paper evaluates O2 on 16 datasets, the diversity of these datasets in terms of domain and characteristics could be expanded. For example, the experiments can have results on some large and standard long-tailed datasets such as ImageNet-LT and CIFAR-LT.
2. The paper could benefit from a more detailed exploration of how these penalty coefficients affect performance and whether they require dataset-specific tuning.
3. The theoretical results are based on certain assumptions that may not hold in all practical scenarios.

**Questions:**

The optional filtering step using a binary classifier is an interesting addition. How does the choice of classifier for this step impact the overall performance of O2? Would different classifiers yield significantly different results?

---

### Official Review · Reviewer_Aqys · 2024-11-02

**Soundness:** 2
**Presentation:** 2
**Contribution:** 1
**Rating:** 1
**Confidence:** 4

**Summary:**

This paper introduces a new oversampling framework, optimized oversampling, which generates synthetic minority class points by maximizing the probability of belonging to the minority class to improve the classifier's performance for imbalanced datasets.

**Strengths:**

1. The problem studied in this paper is important in the literature.
2. The paper is easy to follow.
3. The proposed method is easy to implement.

**Weaknesses:**

1. The proposed method is a data-level approach, and it can be integrated with some classification algorithms such as CART and LR. However, this is very limited in practice as many other popular classifiers are not considered in this paper.
2. The methods compared in the numerical experiments are outdated, and generative model-based methods [1-2] should be considered.
3. The experimental section is slightly weak. The performance of the proposed methods is not significantly better than that of the compared methods, especially for the F1 score, which is the most important metric in imbalanced classification.

[1] Odena, Augustus, et al. "Conditional Image Synthesis with Auxiliary Classifier GANs." 2017

[2] Mariani, Giovanni, et al. "BAGAN: Data Augmentation with Balancing GAN." 2018

**Questions:**

1. Will there be any side effects of the proposed method?
2. Is it possible that the generated synthetic minority class introduce generative error to the model? How to ensure that the generated synthetic minority class are beneficial?

---

### Official Review · Reviewer_5ima · 2024-11-06

**Soundness:** 2
**Presentation:** 3
**Contribution:** 1
**Rating:** 3
**Confidence:** 4

**Summary:**

This paper present an oversampling method O2 by maximizing the probability synthetic point belonging to the minority class. Theoretical guarantees and numerical experimental results are attached.

**Strengths:**

- The experimental results seems to be good.
- The presentation is well.

**Weaknesses:**

The core idea of this paper feels somewhat basic, and there seems to be a gap between its contribution and the typical ICLR standard. The theorems, particularly Theorem 2, appear straightforward and lack deeper insights; for instance, the discrepancy guarantee only presents an inequality without explicit bounds. Additionally, the citation in the main body of the paper is limited, making it challenging to assess the originality of the work; this may be due to the paper's simplicity and a lack of relevant references. Lastly, the experimental results seem limited in significance, though I acknowledge that, as I am less specialized in experimental parts of traditional machine learning, my view here may be less comprehensive—other reviewers may offer a more positive perspective on this aspect.

**Questions:**

N/A

---

### Meta-Review · Area_Chair_Nb19 · 2024-12-21

**Metareview:**

The paper concerns the problem of imbalanced classification. The Authors introduce a new oversampling method that maximizes the probability of belonging to minority class of the generated examples.

The Reviewers identified several flaws in the submission, including a very limited theoretical contribution, outdated references, and insufficient empirical evaluation.

All ratings are below the bar and the Authors did not prepare any rebuttal.

**Additional Comments On Reviewer Discussion:**

There was no discussion as the Authors did not sent rebuttal.

---

### Decision · Program_Chairs · 2025-01-22

Reject